🔬 PLOS | ONE

# Increased BMI has a linear association with late-onset preeclampsia: A population-based study

Pierre-Yves Robillard[1,2]*, Gustaaf Dekker[3], Marco Scioscia[4], Francesco Bonsante[1,2], Silvia Iacobelli[1,2], Malik Boukerrou[2,5], Thomas C. Hulsey[6]

**1** Service de Néonatologie, Centre Hospitalier Universitaire Sud Réunion, Saint-Pierre Cedex, La Réunion, France, **2** Centre d'Etudes Périnatales Océan Indien (CEPOI), Centre Hospitalier Universitaire Sud Réunion, Saint-Pierre Cedex, La Réunion, France, **3** Department of Obstetrics & Gynaecology, University of Adelaide, Robinson Institute, Lyell McEwin Hospital, Adelaide, Australia, **4** Department of Obstetrics and Gynaecology, Policlinico of Abano Terme, Padua, Abano Terme, Italy, **5** Service de Gynécologie et Obstétrique, Centre Hospitalier Universitaire Sud Réunion, Saint-Pierre Cedex, La Réunion, France, **6** Department of Epidemiology, School of Public Health, West Virginia University, Morgantown, WV, United States of America

* pierre-yves.robillard@chu-reunion.fr

**Data Availability Statement:** All relevant data are within the manuscript and its Supporting Information files.

## Abstract

### Background

To investigate the ongoing controversy on the effect of BMI (body mass index) on EOP (early onset preeclampsia) vs LOP (late onset), especially focusing on diabetes and maternal **booking/pre-pregnancy BMI** as possible independent variables.

### Methods

18 year-observational cohort study (2001–2018). The study population consisted of all consecutive births delivered at the Centre Hospitalier Universitaire Hospitalier Sud Reunion's maternity (ap. 4,300 birth per year, only level 3 maternity in the south of Reunion Island, sole allowed to follow and deliver all preeclampsia cases of the area). History of pregnancies, deliveries and neonatal outcomes have been collected in standardized fashion into an epidemiological perinatal data base.

### Results

Chronic hypertension and, history of preeclampsia **in multigravidas**, were the strongest risk factors for EOP. Primiparity, age over 35 years and BMI $\geq$ 35 kg/m$^2$ were rather associated with LOP. In a multivariate analysis with EOP or LOP as outcome variables compared with controls (normotensive), maternal age and pre-pregnancy BMI were independent risk factors for both EOP and LOP (p < 0.001). However, analyzing by increment of 5 (years of age, kg/m$^2$ for BMI) rising maternal ages and incidence of preeclampsia were strictly parallel for EOP and LOP, while increment of BMI was only associated with LOP. Controlling for maternal ages and **booking/pre-pregnancy** BMI, diabetes was not an independent risk factor neither for EOP or LOP.

**Funding:** The author(s) received no specific funding for this work.

**Competing interests:** The authors have declared that no competing interests exist.

## Conclusions

Metabolic factors, other than diabetes, associated with pre-pregnancy maternal corpulence are specifically associated with LOP. This may be a direction for future researches on the maternal preeclamptic syndrome. This may explain the discrepancy we are facing nowadays where high-income countries report 90% of their preeclampsia being LOP, while it is only 60–70% in medium-low income countries.

## Introduction

The recognition of different preeclampsia phenotypes, in particular differentiation between early onset preeclampsia (EOP) and late onset preeclampsia (LOP) has been of pivotal importance in our quest of comprehending the pathophysiology of this still enigmatic syndrome [1,2]. Classical risk factors for preeclampsia may be summarized as: nulliparity, primipaternity, advanced maternal age, previous preeclampsia, high body mass index (BMI), gestational diabetes, multiple gestations, assisted reproduction and pre-existing medical conditions such as chronic hypertension, antiphospholipid antibody syndrome and renal disease [3–5].

Over the past 2 decades, large epidemiologic studies have clearly established that obesity is a major risk for both gestational hypertension and preeclampsia. The risk of preeclampsia typically doubles with each 5–7 kg/m2 increase in pre-pregnancy [6,7]. The mechanisms have only be partially explored; increased cytokine-mediated inflammation and oxidative stress, increased shear stress, dyslipidemia, and increased sympathetic activity1 have all been proposed as possible pathways. An overview of the major processes involved in the pathogenesis of preeclampsia can be referred in reviews [8,9]. It is however less clear whether or not an increasing BMI is associated with all types of PE, in particular EOP and LOP. The data of the 5600 healthy nulliparous pregnant women in the multicentre SCOPE study, just using a binary BMI cut-off of 30, appeared to indicate that obesity was associated with both EOP and LOP, but this paper used a simple binary cut-off of BMI 30 to split the cohort in obese versus non-obese [10].

The problem of maternal corpulence (BMI) is interesting for preeclampsia: it is of note that in the paramount epidemiological efforts made by our masters during the 20th century [11] in big texbooks (Kosmak 1931, Dieckman 1952, Davis 1952, Leon Chesley 1978 & 1999, McGillivray 1983), overweight and obesity were not cited as a problem. Probably in those years obesity was not yet the enormous public health problem on this planet that we are facing nowadays; the real epidemic of obesity has started around the year 1975 [12]. The interest in the effect of obesity as risk factor for preeclampsia seems to have raised in the beginning of 1990's [13], and it became a kind of tsunami since the beginning of the 21st century [6,14–15].

The purpose of the current study involving a large retrospective cohort study (18 year survey, 1736 singleton PE cases) was to resolve the ongoing controversy on the effect of BMI on EOP vs LOP, especially focusing on diabetes and maternal BMI as possible independent variables, and to compare our data with the aforementioned SCOPE study.

## Material and methods

From January 1st, 2001, to 31 December 2018, the hospital records of all women delivered at the maternity of the University South Reunion Island (ap. 4 300 births per year) were abstracted in standardized fashion. The study sample was drawn from the hospital perinatal

database which prospectively records data of all mother-infant pairs since 2001. Information is collected at the time of delivery and at the infant hospital discharge and regularly audited by appropriately trained staff. This epidemiological perinatal data base contains information on obstetrical risk factors, pregnancy complications, description of deliveries and neonatal outcomes. As participants in the French national health care system, all pregnant women in Reunion Island have their prenatal visits, biological and ultasonographic examinations, and anthropological characteristics recorded in their maternity booklet.

Preeclampsia, gestational hypertension and eclampsia were diagnosed according to the definition issued by the International Society for the Study of Hypertension in Pregnancy (ISSHP) relatively to the guidelines in force at the year of pregnancy.

### Design and study population

The maternity department of Saint Pierre hospital is a tertiary care centre that performs about 4 300 deliveries per year, thus representing about 80% of deliveries of the Southern area of Reunion Island, but is the only level 3 maternity (the other maternity is a private clinic, level 1 which is not allowed to follow/deliver preeclamptic pregnancies). Level 1 maternities are allowed to manage pregnancies after 35 weeks of gestation, without severe risk factors (pre-eclampsia, multiple pregnancies, in utero fetal deaths, medical termination of pregnancies, complicated gestational diabetes). Level 2 maternities are allowed to manage deliveries of 33 weeks onward; there are never been level 2 maternities in South-Reunion. Level 3 maternities take care of all pregnancies delivering after 21 weeks gestation and all complicated pregnancies (see level 1 exclusions). Reunion Island is a French overseas region in the Southern Indian Ocean. The entire pregnant population has virtually access to maternity care. This is provided free of charge by the French healthcare system, which combines freedom of medical practice with nationwide social security. Pregnancies are well followed (an average of 9 prenatal visits and 4 ultrasonographies. The mean gestational age at first ("datation") ultrasonography being 11 weeks.

### Definition of exposure and outcomes

The maternal outcomes of interest for this study were obstetric illness (pregnancy-induced hypertension, pre-eclampsia, eclampsia), and way of delivery (normal vaginal delivery, instrumental vaginal delivery—forceps or vacuum—, caesarean section, postpartum haemorrhage,).

Preterm and very preterm birth was defined as delivery before 37 and 33 weeks of gestation, low birth weight (LBW) and very low birth weight (VLBW) as birth weight below 2500 and 1500 g respectively. Infants were considered SGA when the sex and age-adjusted birth weight was below the tenth percentile according to normal tables for our population. Pre-existing renal disorders included glomerulonephritis, reflux- nephropathy and diabetic nephropathy were excluded.

Pre-pregnancy BMI (ppBMI),was calculated on the recalled pre-pregnancy weight by patients themselves in the majority of cases, controlled by the booking weight at first visit (average 6–8 weeks), always written down in the maternity booklet.

### Statistical analysis

Data is presented as numbers and proportions (%) for categorical variables and as mean and standard deviation (SD) for continuous ones. Comparisons between groups were performed by using $\chi^2$-test; odds ratio (OR) with 95% confidence interval (CI) was also calculated. Paired t-test was used for parametric and the Mann-Whitney $U$ test for non-parametric continuous variables. P-values <0.05 were considered statistically significant. Epidemiological data have

been recorded and analysed with the software EPI-INFO 7.1.5 (2008, CDC Atlanta, OMS), EPIDATA 3.0 and EPIDATA Analysis V2.2.2.183. Denmark.

Further, to validate the independent association of maternal pre-pregnancy BMI, or maternal ages and other confounding factors on EOP or LOP we realized a multiple regression logistic model. Variables associated with in bivariate analysis, with a p-value below 0.1 or known to be associated with the outcome in the literature were included in the model. A stepwise backward strategy was then applied to obtain the final model. The goodness of fit was assessed using the Hosmer-Lemeshow test. A p-value below 0.05 was considered significant. All analyses were performed using MedCalc software (version 12.3.0; MedCalc Software's, Ostend, Belgium).

## Ethics approval

This study was conducted in accordance with French legislation. As per new French law applicable to trials involving human subjects (Jardé Act), a specific approval of an ethics committee (comité de protection des personnes- CPP) is not required for this non-interventional study based on retrospective, anonymized data of authorized collections and written patient consent is not needed. This study was registered on UMIN Clinical Trials Registry (identification number is UMIN000037012)

## Results

During the 18-year period, there were 96,861 births in the South of the Reunion Island with an incidence of preeclampsia of 1842 (1.9%), of which 106 multiple pregnancies (5.8%). The study population therefore consisted of 1736 singleton preeclamptic pregnancies (OR for multiple pregnancies as compared with singletons 3.1 [95%CI 2.4–3.9], p< 0.0001).

Table 1 synthesizes all comparisons between EOP and LOP preeclamptic patients (Odds ratios being EOP vs LOP). We put on the left column all results which were not significant. All items statistically significant have been assembled in the right column. EOP women were older than LOP 29.5 vs 28.6 years, p = 0.009, primigravidas (OR 0.78 [0.63–0.96] were prone to LOP. History of abortion or miscarriage were similar in both groups, but history of previous perinatal death or previous preeclampsia was a risk factor for EOP (OR 1.78 [1.2–2.7], p = 0.05 and 1.73 [1.17–2.54], p = 0.001 respectively). EOP women had a tendency to have lower pre-pregnancy BMI (26.4 vs 27.1 kg/m$^2$, p = 0.06), but no significant differences for being overweight (BMI $\geq$ 25 kg/m$^2$) or obese (BMI $\geq$ 30 kg/m$^2$) for the crude results. For these crude results also, Gestational Diabetes Mellitus (GDM) was protective for EOP, OR 0.68 [0.50–0.92], p = 0.01 (but no difference for pre-pregnancy diabetes, p = 0.51). Chronic hypertension (superimposed PE) was a strong factor for EOP, OR 1.47 [1.07–2.0], p = 0.02.

As expected, women who developed EOP had poorer clinical outcomes, with major complications like HELLP syndrome, OR 3.7 [2.5–5.6], p = 0.0001, and placental abruption, OR 6.8 [3.4–14.6] p< 0.001. Neonatal outcome was severely affected by a preterm onset of the disease with intra-uterine fetal deaths 6 times higher than the LOP group (as well as termination of pregnancy for preeclampsia). Perinatal mortality was of 14.3% in EOP vs 1.3% in LOP, OR 12.7 [7.3–22.3], p< 0.0001. Women with LOP had a higher risk of post-partum hemorrhage, OR for EOP 0.30 [0.13–0.7], p = 0.003 (not shown in the Table). No difference was found for eclampsia in the two groups.

Table 2 reports the logistic regression model controlled for maternal age, pre-pregnancy BMI (ppBMI), smoking, gestational diabetes, chronic hypertension and primiparity as compared with controls (women without preeclampsia). Chronic hypertension was the strongest independent risk factor for both EOP and LOP, with an OR of 8.2 [6.1–11.0], p< 0.001and

**Table 1. Crude differences between EOP and LOP.**

| Non significant results<br>Left numbers EOP N = 574<br>Right numbers LOP N = 1162 | P value | Significant results<br>EOP vs LOP<br>ODDS ratios<br>[95% CI] | P value |
|---|---|---|---|
| Gestity (mean, SD)) 2.91 vs 2.73 | 0.10 | Mother Age (years, SD)29.5 vs 28.6 | 0.009 |
| Parity (mean, SD 1.29 vs 1.17 | 0.16 | Primigravidity 0.78 [0.63–0.96] | 0.02 |
| Primiparity 45.3% vs 49.7% OR 0.84 | 0.09 | Chronic hypertension1.47 [1.07–2] | 0.02 |
| Adolescents (<18y) 3.1% vs 3.5% OR 0.89 | 0.67 | Gestational diabetes 0.68 [0.50–0.92] | 0.01 |
| 35 years + 26.1vs 23.8% OR = 1.13 | 0.29 | Atcd perinatal. Deaths1.78 [1.2–2.7] | 0.01 |
| Grand multiparae (5+) 11.0% vs 9.4% OR = 1.19 | 0.29 | Atcd preeclampsia1.73 [1.17–2.5] | 0.01 |
| Single 35.2% vs 38.2% OR = 0.88 | 0.22 | ppBMI* 26.4 vs 27.1 Kg/m$^2$ | 0.06 |
| High school & university56.2% vs 55.1% | 0.70 | | |
| BMI $\geq$ 25 kg/m$^2$53.8% vs 53.6% | 0.93 | | |
| BMI $\geq$ 30 kg/m$^2$ 26.7% vs 30.2%OR = 0.84 | 0.15 | HELLP syndrome 12% vs 3.5% 3.7 [2.5–5.6] | 0.01 |
| Pre-existing diabetes3.9% vs 4.6% | 0.51 | Placental abruption5.6% vs 0.9% 6.8 [3.4–14] | < 0.001 |
| Smoking9.6% vs 8.6% | **0.50** | | |
| Hypercholesterolemia 0.2% vs 0.2% | NS | | |
| Atcd miscarriage30.5% vs 31.0% | 0.84 | Low birthweight <2500g OR = 664 | < 0.001 |
| Atcd abortion26.9% vs 23.0% | 0.14 | Very low birthweight < 1500g 68% vs 1.4%OR = 155 | < 0.001 |
| Atcd thyroid disease#2.4% vs 1.5% OR = 1.52 | 0.21 | SGA * 33.1% vs 25.0%1.47 [1.17–1.85] | < 0.001 |
| Atcd renal disease 1.6% vs 0.9% OR = 1.67 | 0.25 | Perinatal mortality rate 14.3% vs 1.3% 12.7 [7.3–22] | < 0.001 |
| Medically induced preg3.6% vs 2.6% | 0.25 | Medical termination 3.8% vs 1.2%OR = 46.3 | < 0.001 |
| Eclampsia 2.6% vs 2.8% | 0.86 | In utero fetal deaths 6.8% vs 1.2% 6.0 [3.2–11] | < 0.001 |
| Female sex 54.2% vs 51.9% | 0.54 | | |
| Fetal malformations2.3% vs 2.1% | NS | | |

\# goitre, hypo-hyperthyroidy, thyroidectomy, thyroid node, thyroiditis.

* ppBMI pre-pregnancy BMI.

4.95 [3.9–6.3], p <0.001, respectively. Maternal age and pre-pregnancy BMI were independent risk factors for both EOP and LOP. Smoking was protective for LOP (OR 0.74 [0.59–0.91], p = 0.006) but not for EOP. In the multivariate analysis, when controlling for maternal ages and BMI, gestational diabetes was not a risk factor for preeclampsia OR 1.08, p = 0.26, protective for EOP OR 0.73 [0.56–0.98], p = 0.03.

**Table 2. Logistic multivariate regression analysis.** All PE (all preeclampsia N = 1,736), EOP (N = 574) and LOP (N = 1,162) vs controls, no preeclampsia (N = 71,078).

| | All PE<br>aOR | P val | EOP<br>aOR | P val | LOP<br>aOR | P val |
|---|---|---|---|---|---|---|
| Maternal age<br>(increment by 5 years) | 1.04 [1.03–1.05] | < **0.001** | 1.05 [1.03–1.06] | < **0.001** | 1.03 [1.02–1.04] | < **0.001** |
| BMI (kg/m$^2$)<br>(increment by 5kg/m$^2$) | 1.05 [1.04–1.05] | < **0.001** | 1.03 [1.02–1.05] | < **0.001** | 1.05 [1.04–1.06] | < **0.001** |
| Gestational Diabetes | 1.08 | 0.26 | 0.73 [0.56–0.98] | **0.03** | 1.15 | **0.09** |
| Chronic hypertension | 5.62 [4.7–6.8] | < **0.001** | 8.2 [6.1–11.0] | < **0.001** | 4.95 [3.9–6.3] | < **0.001** |
| Tobacco | 0.76 [0.64–0.90] | **0.002** | 0.87 | **0.38** | 0.74 [0.74–0.91] | **0.006** |
| Primiparity | 2.43 [2.2–2.7] | < **0.001** | 2.17 [1.8–2.7] | < **0.001** | 2.44 [2.1–2.8] | < **0.001** |

Women where we could define the pre-pregnancy BMI: N = 965/1162 (83.0%) LOP; N = 456/574 (79.4%) EOP; N = 64,102/71,078 (90.1%) controls.

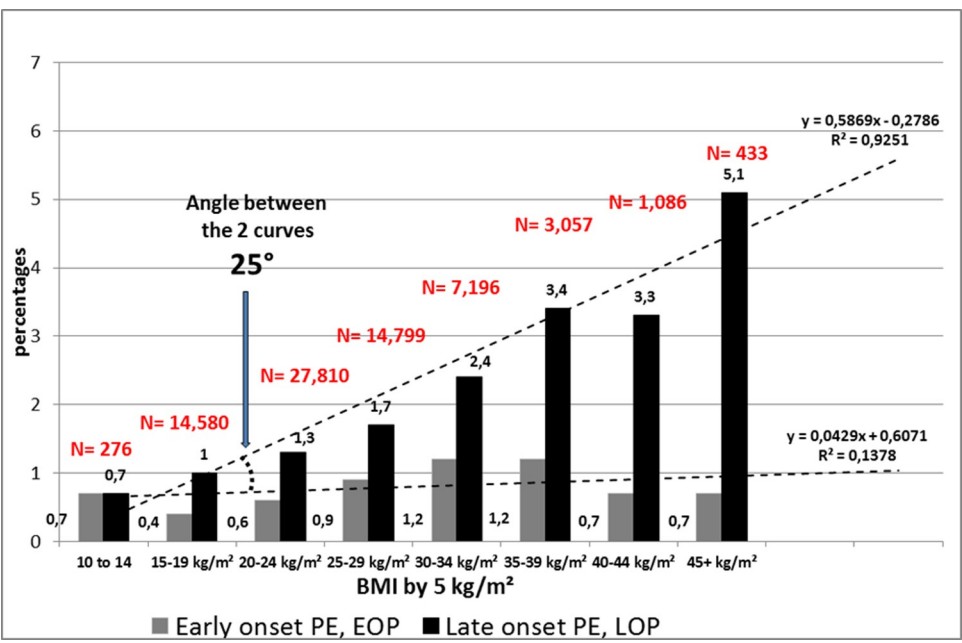

**Fig 1. Prevalence (%) of EOP and LOP per categories of BMI, South Reunion 2001–2018.** 1,736 preeclampsia. 1,587 preeclampsia with BMI, 491 EOP (30.9%) and 1,096 LOP.

Since the logistic regression model demonstrate a strong correlation with ppBMI, we sought to look at the respective incidences of EOP and LOP in Reunion Island (Fig 1) and compared with the international data from the SCOPE study (Fig 2), starting from lean women (<20Kg/m$^2$) to morbid obesity (>40 kg/m$^2$). A linear association was confirmed in

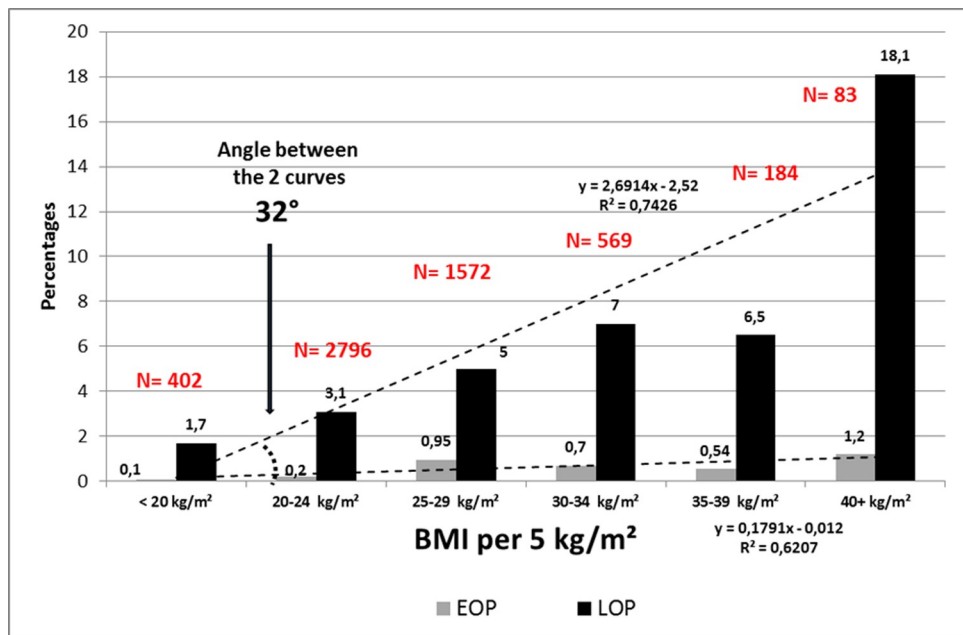

**Fig 2. SCOPE study.** Prevalence (%) of EOP and LOP per categories of BMI,. 278 preeclampsia 28 EOP (10,0%), 250 LOP (34 weeks +).

both groups (EOP and LOP) with a stronger correlation with LOP. Whatever the maternal BMI, the incidence of EOP was relatively constant and almost flat, while there is a linear rise of the incidence of LOP with higher ppBMI. In addition, we wanted to verify this ppBMI effect over time (Fig 3). As a matter of fact, in Reunion island, there is a major public-health problem of a secular rise of obesity. Maternal obesity (ppBMI $\geq$ 30 kg/m$^2$) was approximately of 11–12% in the years 2001–2003 with a positive temporal trend that reached 23% in 2018. Interestingly, the incidence of EOP was quite stable during the study period (ranging between 0.7 and 1.1‰) while LOP seems to follow closely the obesity trend doubling in the study period (from 12‰ in 2001 to 24‰ in 2018).

Another maternal characteristic that was revealed by the logistic regression model was a correlation between maternal age and the development of preeclampsia (p<0.001) so we plotted the incidence of preeclampsia according to this variable (Fig 4). There is a regular rise of EOP and LOP with maternal ages, describing two parallel lines of EOP and LOP. Also Table 3 shows the bivariate interaction between maternal ages and BMI both together. Adjusted Odds ratios for maternal ages and BMI are similar in EOP women (1.035 [1.02–1.05], 1.039 [1.03–1.05] respectively). This is not the case for LOP: aOR for maternal ages is 1.015 [1.006–1.024] while it raises at 1.056 [1.05–1.06] for maternal pre-pregnancy BMI. In all preeclamptic women, there is a higher increment for BMI, aOR 1.05 (i.e. an increment of 5% for each increment of 5kg/m$^2$) than for maternal ages aOR 1.03 (an increment of 3% for each increment of 5 years of age). But this effect is concentrated only in LOP. Figs 1 and 2 and 4 show geometrically that indeed, in one case (pre-pregnancy BMI) the two curves diverge by an angle of ap. 30˚, while it is a parallel rise when we consider maternal ages.

## Discussion

The results of this large retrospective cohort clearly demonstrate that (a) chronic hypertension is the strongest risk factors for preeclampsia, but mainly linked with EOP, b) nulliparity at

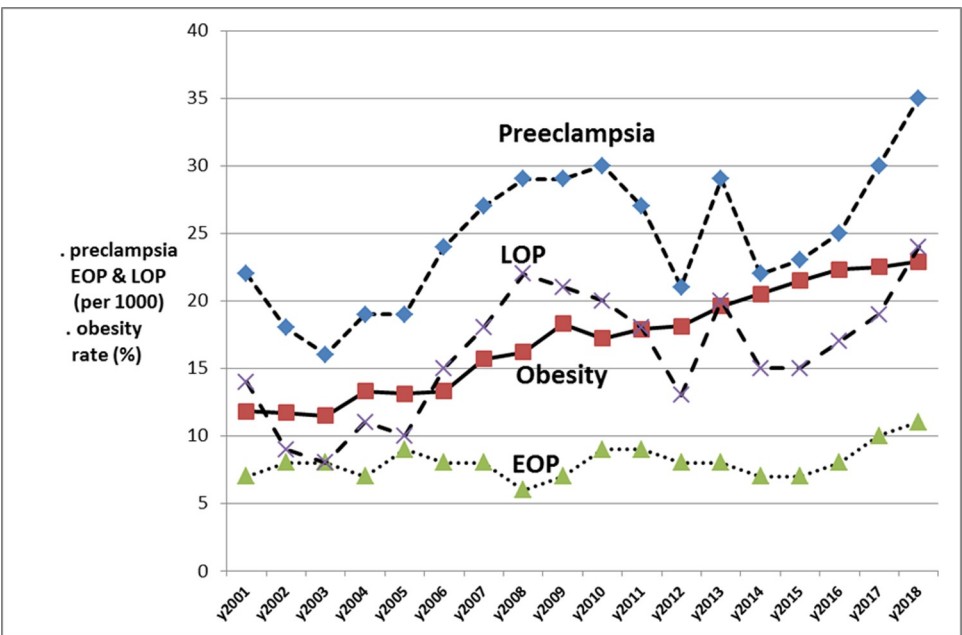

**Fig 3. Evolution per year 2001–2018 in South-Réunion of the incidence of obesity (in per cents), and the incidence of preeclampsia, early onset preeclampsia and late onset preeclampsia (per 1000).**

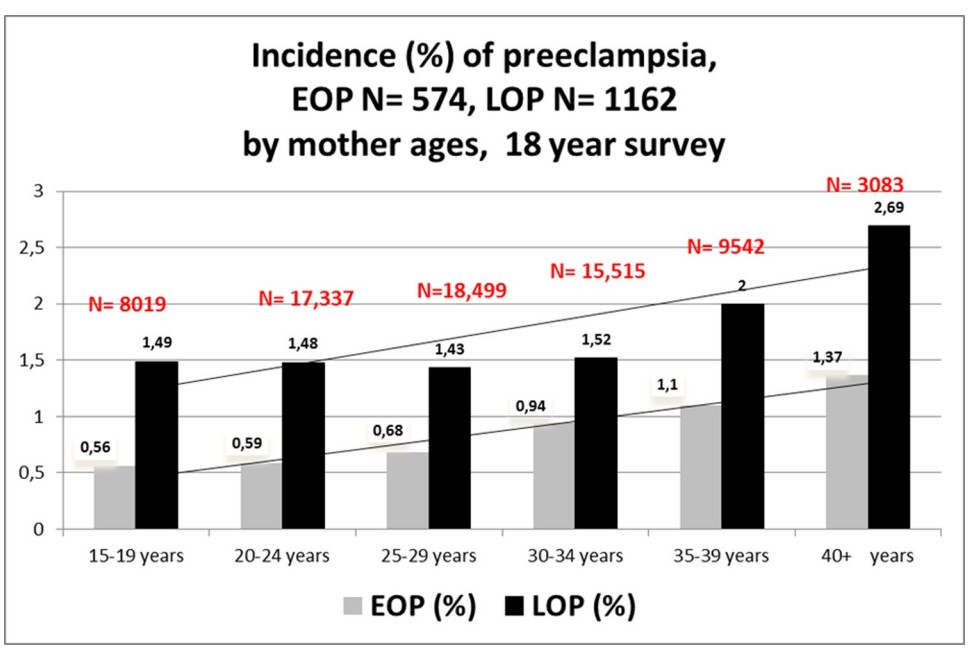

**Fig 4. Incidence of preeclampsia, EOP (N = 574), LOP (N = 1,162) by mother ages.**

booking is linked rather with LOP than with EOP, although OR in the multivariate analysis are similar (Table 2), c) previous preeclampsia is strongly linked with EOP, e) pre-existing medical conditions such as thyroid disease are linked with both EOP and LOP, f) advanced maternal age is both linked with the occurrence of EOP and LOP, but in a paralleled and coupled rise (see Table 3 and Fig 4) f) pre-pregnancy maternal BMI is also linked with the EOP and LOP, but the paralleling and coupling is with LOP (see Figs 1–3) and no clear correlation with EOP, and finally, g) gestational diabetes is not an independent risk factor for preeclampsia when controlling for maternal ages and maternal pre-pregnancy BMI (ppBMI). Pre-conceptional weight and weight gain during pregnancy were found to be associated with pregnancy complications in a recent meta-analysis of European, North American and Australian cohorts although the correlation between preeclampsia and ppBMI was not investigated in detail [16]. Furthermore, pre-existing diabetes is a well-established risk for diabetes, and several studies previously proposed GDM as a risk factor (risk X by 2 or 4 [17]). Reports associate diabetes and preeclampsia [18–20], especially with LOP [20]. However, a few studies [21,22] do not find such an association when controlling for maternal weight and age, as in a very recent study, in a cohort of 15,000 pregnant women in Beijing, China [23]. In our

**Table 3. Logistic bivariate regression analysis between maternal ages and pre-pregnancy BMI.** All PE (all preeclampsia N = 1,736), EOP (N = 574) and LOP (N = 1,162) vs controls, no preeclampsia (N = 71,078).

| | All PE aOR | P val | EOP aOR | P val | LOP aOR | P val |
|---|---|---|---|---|---|---|
| **Maternal age (increment by 5 years)** | 1.021 [1.04–1.029] | < **0.001** | 1.035 [1.021–1.048] | < **0.001** | 1.015 [1.006–1.024] | < **0.001** |
| **BMI (kg/m²) (increment by 5kg/m²)** | 1.050 [1.043–1.057] | < **0.001** | 1.039 [1.026–1.053] | < **0.001** | 1.056 [1.047–1.064] | < **0.001** |

Women where we could define the pre-pregnancy BMI: N = 965/1162 (83.0%) LOP; N = 456/574 (79.4%) EOP; N = 64,102/71,078 (90.1%) controls.

prospective cohort also, after controlling for maternal pre-gestational BMI and maternal ages, gestational diabetes is no longer an independent risk for preeclampsia, and especially for late onset PE. These findings suggest that a certain degree of mild hyperglycaemia per se does not adversely affect maternal-placental homeostasis.

In our experience, growth retardation in newborns is higher in EOP pregnancies (33% of newborns), but remains relatively high (25%) in LOP infants (OR 1.47, [1.17–1.85], p = 0.001 for EOP, Table 1), which was not the case in the SCOPE study (LOP babies were mainly normal weighted) [10]. This 18-year cohort confirms that chronic hypertension and previous history of preeclampsia are paramount risks for PE and notably EOP. Prior preeclampsia and chronic hypertension have been ranked first and second in a meta analysis recalling some 25 million pregnancies [24]. However, it should be noted that while in this cohort 178/1736 (10.3%) women had PE superimposed on chronic hypertension, and 131 (7.6%) preeclamptic multiparous women had a preeclampsia in an earlier pregnancy (representing 15% of our preeclamptic multiparae [25]), 1459 (84%) of our preeclamptics did not present any of these two risks, highlighting the major importance of other major risk factors. Advancing maternal age is usually assumed as a risk factor for preeclampsia [22,26–27], while some studies contest that advanced age is involved [14, 28–29]. In this study, there is a rise in preeclampsia incidence with maternal ages (Fig 4), but it is exactly the same parallel rise for EOP and LOP.

It is important to note that the distinction EOP vs LOP, albeit important' is arbitrary. Over the past decade the pivotal role of syncytiotrophoblast (STB) stress [30–32] and the resulting anti-angiogenic state have been recognised as the 'final common pathway' to endothelial dysfunction and as such the maternal syndrome PE. Poor spiral artery remodelling is typically associated with IUGR [33] and therefore EOP. Lack of spiral artery remodelling with subsequent STB damage by damaging 'pulsatile' flows [34] and subsequent ischaemia reperfusion event clearly represent an important pathway leading to STB stress. One can easily hypothesize why chronic hypertension with the stiffer, thicker arterial walls will lead to EOP. The biology of adverse factors like maternal age per se are not directly evident, STB stress leading to PE has also been proposed as reflecting premature STB aging. Would advanced maternal age associate with premature STB aging? LOP is mostly not associated with inadequate spiral artery remodelling, other factors are required to trigger STB stress, with BMI clearly representing a very important player, both the status of chronic inflammation as increased triglyceride levels but also increased sympathetic stress [35] being possible adverse vectors.

High BMI, especially obesity, increase the risk of preeclampsia as extensively documented in multiple studies [6,15,36]. But several studies investigating preeclampsia subtypes (EOP and LOP) reported specifically that being overweight/obese increased the risk of LOP and not the risk of EOP [14,23, 37–39]. The SCOPE consortium had reported that obesity was both a similar risk factor for EOP and LOP [10], but using a binary definition of obesity (30+/-, yes/no). In this study, we demonstrate that using the BMI as a continuum, we find in both cases that increasing BMI is linearly associated with the incidence of LOP. Further, we wanted to double check this BMI effect over time: as a matter of fact, in Reunion island, we have a major public-health problem of a secular rise of obesity ($\geq$ 30 kg/m$^2$). As epidemiologists, both obstetricians and neonatologists, we witness with some helplessness this regular rise of obesity year after year in our context. Interestingly, the preeclampsia incidence is quite a twin curve of that of LOP during these 18 years (Fig 3).

Obesity has been shown to represent a definite risk for the development of preeclampsia during pregnancy [40]. Obese patients present a metabolic syndrome (MS) characterized by insulin resistance with excessive flux of fatty acids and a proinflammatory state that contribute to disrupt the equilibrium between immunology, metabolic alterations and systemic inflammation that are necessary for a healthy pregnancy.

The SCOPE study demonstrated that all variables making up the metabolic syndrome, so not only obesity, are risk factors for PE [41]. In clinically evident preeclampsia, an exacerbated MS occurs, and this may represent the major risk of cardiovascular and metabolic disease in later life [42,43]. In fact, a close relationship between hypertension, endothelial and kidney damage (features shared between preeclampsia and MS) hold a pivotal role for long term complications [43,44] as demonstrated by interventions to improve insulin sensitivity that reduce the risk of both hypertension in pregnancy and later life cardiovascular complications [45,46]. Metabolic and inflammatory intracellular pathways share some mediators like the insulin receptor substrate, phosphatidyl inositol 3-kinase, and the protein kinase B [47] that are also involved in the immunological activation of natural killer cells [48] and in the angiogenic imbalance that was demonstrated in preeclampsia [49,50]. The association between the MS and preeclampsia is supported by evidence of increased hyperinsulinemia [48], abnormal placental accumulation of glycogen [51] and unusual insulin mediators [52] with subsequent impaired placental insulin signaling. Certainly, insulin resistance act synergistically with impaired angiogenic factors [49,50] that lead to the preeclamptic syndrome (hypertension, proteinuria, and organ damage) [53–54].

A strength of this study is the fact that it represents a true population study. The Centre Hospitalier Universitaire Sud-Reunion's maternity (Level 3, European standards of care) is the only public hospital in the southern part of Reunion Island (Indian Ocean, French overseas department). It serves the whole population of the area (ap. 360,000 inhabitants, and 5100 births per year). With 4,300 births per year, the university maternity represents 82% of all births in the South. But, as a level 3 (the other maternity is a private clinic, level 1), we are sure to have had ALL the preeclampsia of the south of the island during the 18 year period. During all the 18-year period, definitions of preeclampsia and HELLP syndrome have been the same. The weakness of this study is that the available data base does not provide data on length of sexual relationship and/or primipaternity.

## Conclusions

"Getting rid" of diabetes in preeclampsia, while recognising that late onset preeclampsia is linearly associated with rising maternal pre-pregnancy body mass index (and not early onset PE) may narrow the beam toward future researches on carbohydrate/lipid metabolism explaining the maternal inflammation syndrome at the 3rd trimester of pregnancy. This problem of corpulence may also explain the discrepancy we are facing nowadays on this planet where high-income countries report 90% of their preeclampsia being LOP [55], while it is only 60–70% in medium-low income countries (unfortunately, still the great majority of humankind) [56] as well as the reports of the 20th century's scholars [57,58], where, for example, our masters did not emphasized at all obesity.

## Supporting information

**S1 File. EOP BMI: Calculations of BMI and EOP in the whole population.**
(DOC)

**S2 File. LOP BMI: Calculations of BMI and LOP in the whole population.**
(DOC)

**S3 File. EOP vs LOP 19: Main calculations EOP vs LOP.**
(DOC)

**S4 File. EOP vs LOP singletons: Complementary calculations EOP vs LOP.**
(DOCX)

**S5 File. Logistic regression age BMI: Calculations of logistic regression.**
(DOCX)

**S6 File. Logistic regression EOP LOP: Cal culations logistic regression.**
(DOCX)

## Author Contributions

**Conceptualization:** Pierre-Yves Robillard, Malik Boukerrou.

**Data curation:** Pierre-Yves Robillard, Malik Boukerrou.

**Formal analysis:** Gustaaf Dekker, Marco Scioscia, Francesco Bonsante, Thomas C. Hulsey.

**Investigation:** Pierre-Yves Robillard.

**Methodology:** Pierre-Yves Robillard, Thomas C. Hulsey.

**Software:** Francesco Bonsante.

**Supervision:** Gustaaf Dekker, Marco Scioscia, Silvia Iacobelli, Malik Boukerrou, Thomas C. Hulsey.

**Validation:** Gustaaf Dekker, Marco Scioscia, Francesco Bonsante, Silvia Iacobelli, Thomas C. Hulsey.

**Visualization:** Silvia Iacobelli, Thomas C. Hulsey.

**Writing – original draft:** Pierre-Yves Robillard, Gustaaf Dekker, Marco Scioscia, Thomas C. Hulsey.

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
