## [Decision Letter · Decision Letter 0]

29 Aug 2019

[EXSCINDED]

PONE-D-19-21680

LATE ONSET PREECLAMPSIA IS LINEARLY ASSOCIATED WITH RISING MATERNAL PRE-PREGNANCY BMI. A POPULATION-BASED STUDY of 1,700 CASES of PREECLAMPSIA IN SINGLETON PREGNANCIES.

PLOS ONE

Dear Dr. Robillard,

Thank you for submitting your manuscript to PLOS ONE. After careful consideration, we feel that it has merit but does not fully meet PLOS ONE’s publication criteria as it currently stands. Therefore, we invite you to submit a revised version of the manuscript that addresses the points raised during the review process.

We would appreciate receiving your revised manuscript by 10/27/2019. To enhance the reproducibility of your results, we recommend that if applicable you deposit your laboratory protocols in protocols.io, where a protocol can be assigned its own identifier (DOI) such that it can be cited independently in the future. For instructions see: http://journals.plos.org/plosone/s/submission-guidelines#loc-laboratory-protocols

We look forward to receiving your revised manuscript.

Kind regards,

Linglin Xie

Academic Editor

PLOS ONE

Journal Requirements:

4. Please upload a copy of Figure 5, to which you refer in your text on page 8. If the figure is no longer to be included as part of the submission please remove all reference to it within the text.

Reviewers' comments:

Reviewer's Responses to Questions

**Comments to the Author**

1. Is the manuscript technically sound, and do the data support the conclusions?

Reviewer #1: Yes

Reviewer #2: Partly

2. Has the statistical analysis been performed appropriately and rigorously? 

Reviewer #1: Yes

Reviewer #2: Yes

3. Have the authors made all data underlying the findings in their manuscript fully available?

Reviewer #1: Yes

Reviewer #2: No

4. Is the manuscript presented in an intelligible fashion and written in standard English?

Reviewer #1: Yes

Reviewer #2: Yes

5. Review Comments to the Author

Reviewer #1: In this paper, authors did a 18 year-observational cohort study to analyse the parameters related with EOP and LOP and compared their different effects on these two preeclampsia. It is a relatively comprehensive research based on so large number of cases. And authors got the interesting findings that increment of 5 years for maternal age rising and incidence of preeclampsia were strictly parallel for EOP and LOP, while increment of 5kg/m² for BMI was only associated with LOP. That can give us some clues on the different mechanism of EOP and LOP.

There are some problems need to be modified:

1, In the method, authors quoted the levels of maternity, like level 1, level 3, but didn't give some background of the definition of the levels for maternity. They need introduce the division of levels.

2, In the findings, authors found that increment of 5 (years for age, 5kg/m² for BMI) rising maternal ages and incidence of preeclampsia were strictly parallel for EOP and LOP, while increment of BMI was only associated with LOP. It is hard to understand when put 5kg/m² for BMI with 5 years for age to explain the relationship between maternal ages and incidence of preeclampsia. It would be better if they put the information of 5kg/m² for BMI down when concluded that BMI was only associated with LOP.

3, Some abbreviations, like LDW, VLDW, GDM, ppBMI need to write their full names when first using them in this manuscript. And authors should make the whole paper be more organised.

Reviewer #2: Pierre-Yves Robillard et al. investigated the effect of BMI (body mass index) on EOP (early onset preeclampsia) vs LOP (late onset) by a long term observational cohort study. And they found that increment of maternal BMI was only associated with LOP during pregnancy. This was a great job. However, there were some issues in the manuscript.

And the suggestion of the manuscript followed:

1. The authors did not highlight the significance of BMI for preeclampsia such as EOP or LOP in the introduction, and there was some confusions in the logical structure in the part;

2. The maternal BMI analysis of the cohort study was not enough in the Results. It was suggested to set a time point for maternal BMI before and during pregnancy.

3. The format of Table 4 needs to be revised to meet the requirements of article publication; Table 1-2 has too many entries and needs to be re-integrated and revised;

4. Figures 1-4 should be illustrated with brief annotations;

5. The limitations of the paper need to be explained in detail;

6. The conclusion of the paper did not match the title, so the conclusion should be written around the center of the paper.

6. PLOS authors have the option to publish the peer review history of their article (what does this mean?). If published, this will include your full peer review and any attached files.

Reviewer #1: No

Reviewer #2: No

---

## [Author Response · Author response to Decision Letter 0]

12 Sep 2019

[PONE-D-19-21680] - [EMID:49f54e1ae9a6df0f]

After comments of the reviewers, we have shortened the title which is now:

 Increased BMI has a linear association with late-onset 

 preeclampsia: a population-based study

Verified

We did not understand at first, we will send all the anonymized calculations

Figures will be added separately in TIFF format

4. Please upload a copy of Figure 5, to which you refer in your text on page 8. If the figure is no longer to be included as part of the submission please remove all reference to it within the text.

It was a typing mistake, in fact it is Figure 4, corrected

5. Review Comments to the Author

Reviewer #1: In this paper, authors did a 18 year-observational cohort study to analyse the parameters related with EOP and LOP and compared their different effects on these two preeclampsia. It is a relatively comprehensive research based on so large number of cases. And authors got the interesting findings that increment of 5 years for maternal age rising and incidence of preeclampsia were strictly parallel for EOP and LOP, while increment of 5kg/m² for BMI was only associated with LOP. That can give us some clues on the different mechanism of EOP and LOP.

There are some problems need to be modified:

1, In the method, authors quoted the levels of maternity, like level 1, level 3, but didn't give some background of the definition of the levels for maternity. They need introduce the division of levels.

Sentences have been added page 4 (in blue in the text):

“Level 1 maternities are allowed to manage pregnancies after 35 weeks of gestation, without severe risk factors (preeclampsia, multiple pregnancies, in utero fetal deaths, medical termination of pregnancies, complicated gestational diabetes). Level 2 maternities are allowed to manage deliveries of 33 weeks onward; there are never been level 2 maternities in South-Reunion. Level 3 maternities take care of all pregnancies delivering after 21 weeks gestation and all complicated pregnancies (see level 1 exclusions).

Pregnancies are well followed (an average of 9 prenatal visits and 4 ultrasonographies. The mean gestational age at first (“datation”) ultrasonography being 11 weeks.” 

2, In the findings, authors found that increment of 5 (years for age, 5kg/m² for BMI) rising maternal ages and incidence of preeclampsia were strictly parallel for EOP and LOP, while increment of BMI was only associated with LOP. It is hard to understand when put 5kg/m² for BMI with 5 years for age to explain the relationship between maternal ages and incidence of preeclampsia. It would be better if they put the information of 5kg/m² for BMI down when concluded that BMI was only associated with LOP.

Increments by 5 years of age, and by 5 kg/m² indeed look similar on the graphs and therefore may be confusing. We have added in the different figures clearly in abscissa years of ages or kg/m².The statistical comparison using 5-unit intervals (BMI and years of age) allows to evaluate more homogeneous data than continuous variables with lower intergroup variability. 

 Further, we have added in Table 3 a Logistic bivariate regression analysis between maternal ages and pre-pregnancy BMI showing their respective interactions. We have added, page 7 the sentence:

 “Also Table 3 shows the bivariate interaction between maternal ages and BMI both together. Adjusted Odds ratios for maternal ages and BMI are similar in EOP women (1.035 [1.02-1.05], 1.039 [1.03-1.05] respectively. This is not the case for LOP: aOR for maternal ages is 1.015 [1.006-1.024] while it raises at 1.056 [1.05-1.06] for maternal pre-pregnancy BMI. In all preeclamptic women, there is a higher increment for BMI, aOR 1.05 (i.e. an increment of 5% for each increment of 5kg/m²) than for maternal ages aOR 1.03 (an increment of 3% for each increment of 5 years of age). But this effect is concentrated only in LOP. Fig 1-2 and Fig 4 show geometrically that indeed, in one case (pre-pregnancy BMI) the two curves diverge by an angle of ap. 30°, while it is a parallel rise when we consider maternal ages”.

3, Some abbreviations, like LDW, VLDW, GDM, ppBMI need to write their full names when first using them in this manuscript: corrected page 4

 And authors should make the whole paper be more organised. 

The introduction and the conclusions have been completely re-written (in blue in the revised text)

 

Reviewer #2: Pierre-Yves Robillard et al. investigated the effect of BMI (body mass index) on EOP (early onset preeclampsia) vs LOP (late onset) by a long term observational cohort study. And they found that increment of maternal BMI was only associated with LOP during pregnancy. This was a great job. However, there were some issues in the manuscript.

And the suggestion of the manuscript followed:

1. The authors did not highlight the significance of BMI for preeclampsia such as EOP or LOP in the introduction, and there was some confusions in the logical structure in the part;

The introduction and the conclusions have been completely re-written (in blue in the revised text)

2. The maternal BMI analysis of the cohort study was not enough in the Results. It was suggested to set a time point for maternal BMI before and during pregnancy.

All data are pre-pregnancy BMI. The study focuses on pre-pregnancy BMI only, not on weight gain during pregnancy. We have added in material and methods, page 5 the sentence:

“Pre-pregnancy BMI (ppBMI),was calculated on the recalled pre-pregnancy weight by patients themselves in the majority of cases, controlled by the booking weight at first visit (average 6-8 weeks), always written down in the maternity booklet.”

3. The format of Table 4 needs to be revised to meet the requirements of article publication; Table 1-2 has too many entries and needs to be re-integrated and revised

- Table 1-3 have been deleted and replaced by a synthetic Table which is now Table 1.

- Table 4 (logistic regression analysis), now Table 2 and 3 have been revised to meet the requirements

Figures 1-4 should be illustrated with brief annotations. 

We have added in the different figures clearly in abscissa years of ages or kg/m².

5. The limitations of the paper need to be explained in detail; 

The paragraph is the last one of page 10 (in blue in the text)

6. The conclusion of the paper did not match the title, so the conclusion should be written around the center of the paper.

THE TITLE has been highly shortened and is now:

Increased BMI has a linear association with late-onset preeclampsia: a population-based study 

The introduction and the conclusions have been completely re-written (in blue in the revised text)

---

## [Decision Letter · Decision Letter 1]

25 Sep 2019

PONE-D-19-21680R1

Increased BMI has a linear association with late-onset 

                   preeclampsia: a population-based study

PLOS ONE

Dear Dr. Robillard,

Thank you for submitting your manuscript to PLOS ONE. After careful consideration, we feel that it has merit but does not fully meet PLOS ONE’s publication criteria as it currently stands. Therefore, we invite you to submit a revised version of the manuscript that addresses the points raised during the review process.

We would appreciate receiving your revised manuscript by 10/25/2019. To enhance the reproducibility of your results, we recommend that if applicable you deposit your laboratory protocols in protocols.io, where a protocol can be assigned its own identifier (DOI) such that it can be cited independently in the future. For instructions see: http://journals.plos.org/plosone/s/submission-guidelines#loc-laboratory-protocols

We look forward to receiving your revised manuscript.

Kind regards,

Linglin Xie

Academic Editor

PLOS ONE

Reviewers' comments:

Reviewer's Responses to Questions

**Comments to the Author**

1. If the authors have adequately addressed your comments raised in a previous round of review and you feel that this manuscript is now acceptable for publication, you may indicate that here to bypass the “Comments to the Author” section, enter your conflict of interest statement in the “Confidential to Editor” section, and submit your "Accept" recommendation.

Reviewer #1: All comments have been addressed

Reviewer #2: (No Response)

2. Is the manuscript technically sound, and do the data support the conclusions?

Reviewer #1: Yes

Reviewer #2: Yes

3. Has the statistical analysis been performed appropriately and rigorously? 

Reviewer #1: Yes

Reviewer #2: Yes

4. Have the authors made all data underlying the findings in their manuscript fully available?

Reviewer #1: Yes

Reviewer #2: Yes

5. Is the manuscript presented in an intelligible fashion and written in standard English?

Reviewer #1: Yes

Reviewer #2: Yes

6. Review Comments to the Author

Reviewer #1: This article analyzed the relationship between BMI and LOP or EOP. And can give us some useful information from the conclusion.

Reviewer #2: Although the tables had been revised according to the reviewer's opinions, there are still some problems in the format, which do not meet the requirements of publication. It is suggested that the author read the form requirements of this journal carefully.

7. PLOS authors have the option to publish the peer review history of their article (what does this mean?). If published, this will include your full peer review and any attached files.

Reviewer #1: No

Reviewer #2: No

---

## [Author Response · Author response to Decision Letter 1]

27 Sep 2019

Reviewers' comments:

Reviewer's Responses to Questions

Comments to the Author

1. If the authors have adequately addressed your comments raised in a previous round of review and you feel that this manuscript is now acceptable for publication, you may indicate that here to bypass the “Comments to the Author” section, enter your conflict of interest statement in the “Confidential to Editor” section, and submit your "Accept" recommendation.

Reviewer #1: All comments have been addressed

Reviewer #2: (No Response)

2. Is the manuscript technically sound, and do the data support the conclusions?

Reviewer #1: Yes

Reviewer #2: Yes

3. Has the statistical analysis been performed appropriately and rigorously?

Reviewer #1: Yes

Reviewer #2: Yes

4. Have the authors made all data underlying the findings in their manuscript fully available?

Reviewer #1: Yes

Reviewer #2: Yes

5. Is the manuscript presented in an intelligible fashion and written in standard English?

Reviewer #1: Yes

Reviewer #2: Yes

6. Review Comments to the Author

Reviewer #1: This article analyzed the relationship between BMI and LOP or EOP. And can give us some useful information from the conclusion.

Reviewer #2: Although the tables had been revised according to the reviewer's opinions, there are still some problems in the format, which do not meet the requirements of publication. It is suggested that the author read the form requirements of this journal carefully.

We have carefully made changes in Table 1 and Table 2 (in red in the manuscript track changes). We hope that the modifications fit with the Plos requirements

7. PLOS authors have the option to publish the peer review history of their article (what does this mean?). If published, this will include your full peer review and any attached files.

Do you want your identity to be public for this peer review? For information about this choice, including consent withdrawal, please see our Privacy Policy.

Reviewer #1: No

Reviewer #2: No

---

## [Editor Report · Decision Letter 2]

2 Oct 2019

Increased BMI has a linear association with late-onset 

                   preeclampsia: a population-based study

PONE-D-19-21680R2

Dear Dr. Robillard,

We are pleased to inform you that your manuscript has been judged scientifically suitable for publication and will be formally accepted for publication once it complies with all outstanding technical requirements.

With kind regards,

Linglin Xie

Academic Editor

PLOS ONE
---

## [Editor Report · Acceptance letter]

8 Oct 2019

PONE-D-19-21680R2 

Increased BMI has a linear association with late-onset preeclampsia: a population-based study

Dear Dr. Robillard:

I am pleased to inform you that your manuscript has been deemed suitable for publication in PLOS ONE. Congratulations! Your manuscript is now with our production department. 

With kind regards,

on behalf of

Dr. Linglin Xie 

Academic Editor

PLOS ONE